# On Fair and Balanced Matching in Bipartite Graphs

## Abstract

Two-sided matching in bipartite graphs under fairness constraints is a fundamental problem with applications in admissions, hiring, and resource allocation. Recent work has shown that neural architectures such as WeaveNet can approximate stable matchings with high fidelity. In this paper, we extend WeaveNet's results by directly training on their proposed datasets, and introducing a new loss function taking into consideration the positions in the ranking. We then compare WeaveNet with graph convolutional networks (GCNs), graph neural networks (GNN) and graph attention networks (GATs) architectures proposed by us, explicitly incorporating attention mechanisms to capture preference-aware dependencies. Beyond stability, we evaluate matchings using fairness-aware metrics, including envy-freeness up to $k$ peers (EF-$k$), weighted logarithmic rank (WLR), exponential score difference (ESD), and average rank difference (ARD). Our results highlight both the strengths and limitations of WeaveNet compared to attention-based GNN and GCN models, suggesting that hyperparameter-induced biases can dominate architectural design choices in fairness-aware optimization.

## 1   Introduction

Two-sided matching problems arise in many real-world settings, including student-university admissions, organ donation exchanges, and online marketplaces. Classical approaches such as the Gale–Shapley deferred acceptance algorithm Gale and Shapley [1962] guarantee stability, ensuring no blocking pairs exist, but systematically favor the proposing side and may lead to imbalanced or unfair outcomes.

Recent work has highlighted the importance of fairness criteria, such as envy-freeness, sex-equality, and balance, to better reflect equitable outcomes Cho et al. [2024], Coutance et al. [2023]. However, algorithmic solutions to fairness-aware matching often face computational hardness or limited scalability . Parallel to these developments, graph neural networks (GNNs), graph convolutional networks (GCN) and graph attention networks (GAT) have shown promise for combinatorial optimization tasks Li [2019], Gibbons et al. [2019]. In particular, WeaveNet Sone et al. [2023b] demonstrated that specialized architectures can approximate stable matchings with high fidelity.

In this paper, we reproduce and extend the WeaveNet framework by (i) training on datasets proposed by their paper, (ii) evaluating fairness-aware criteria including envy-freeness up to $k$ (EF-$k$), Weighted Logarithmic Rank (WLR), Average Rank Difference (ARD) and Exponential Score Difference (ESD), and (iii) comparing WeaveNet against graph neural networks (GNNs), graph convolutional networks (GCNs) and graph attention networks (GATs) that incorporate explicit attention mechanisms. Our results demonstrate that while GCNs achieve perfect stability, fairness objectives require careful loss weight calibration to influence training.

## 2   Background and Related Work

Matching is the task of allocating agents on one side of a bipartite graph to agents on the other, under constraints such as preferences and capacities. Classical solutions include the Gale–Shapley deferred acceptance (DA) algorithm Gale and Shapley [1962], which guarantees stable outcomes but can be biased toward the proposing side. Extensions such as Deferred Acceptance with Compensation Chains (DACC) Dworczak [2016], PolyMin , and PowerBalance  aim to mitigate these trade-offs by incorporating fairness objectives or balancing mechanisms.

WeaveNet Sone et al. [2023b] is a neural architecture specifically designed for bipartite matching. It introduces a two-stream feature weaving mechanism that preserves edge-level information while avoiding over-smoothing in dense graphs. WeaveNet demonstrated competitive performance compared to algorithmic baselines on small matching instances and provided the first unsupervised GNN-based framework for stable matchings. However, WeaveNet primarily optimized for stability and sex-equality costs, leaving open the question of how alternative architectures (e.g., GNNs with attention) and broader fairness metrics such as EF-$k$ and WLR perform. This motivates our comparative study.

### 2.1   Preliminaries

**Envy-Freeness (Cho et al. [2024]) up to $k$ peers:** Let $Y$ be a matching and let $s \in S$ be a student. Define the set of students that $s$ has justified envy toward in $Y$ as:

$$\text{Ev}(Y, s) = \{s' \in S \mid s \text{ has justified envy toward } s' \text{ in } Y\}.$$

The matching $Y$ is said to be *envy-free up to $k$ peers* (EF-$k$) if: $\forall s \in S, \quad |\text{Ev}(Y, s)| \leq k$. In particular, EF-0 corresponds to full fairness (no student envies another), while any matching trivially satisfies EF-$(n-1)$, where $n = |S|$ is the total number of students.

**Satisfaction of both sides:** Let $S^A \in (0, 1]^{N \times M}$ and $S^B \in (0, 1]^{M \times N}$ denote the normalized (following WeaveNet's normalization Sone et al. [2023b] ) preference scores of agents in groups $A$ and $B$, respectively, where larger values correspond to higher preference. For a predicted matching $\hat{m} \in [0, 1]^{N \times M}$, the satisfaction of each side is defined as:

$$S(\hat{m}; A) = \sum_{i=1}^{N} \sum_{j=1}^{M} \hat{m}_{ij} \cdot S^A_{ij}, \quad S(\hat{m}; B) = \sum_{j=1}^{M} \sum_{i=1}^{N} \hat{m}_{ij} \cdot S^B_{ji} \tag{1}$$

## 3   Method: Fair and Balanced Matching in Bipartite Graphs

The **absolute difference** formulation, used in the original WeaveNet paper, applies a linear penalty:

$$\mathcal{L}^{\text{abs}}_f(\hat{m}; I) = \frac{1}{N} |S(\hat{m}; A) - S(\hat{m}; B)| \tag{2}$$

To explore different penalty structures for side imbalance, we evaluate three variants of the fairness loss $\mathcal{L}_f$. The **squared difference** formulation penalizes imbalances quadratically:

$$\mathcal{L}^{\text{sq}}_f(\hat{m}; I) = \frac{1}{N^2} (S(\hat{m}; A) - S(\hat{m}; B))^2 \tag{3}$$

To test whether exponentially penalizing large imbalances improves fairness, we propose a **log-exponential** variant that applies a smooth exponential penalty while maintaining numerical stability:

$$\mathcal{L}^{\text{exp}}_f(\hat{m}; I) = \log\left(1 + \exp\left(\frac{|S(\hat{m}; A) - S(\hat{m}; B)|}{N}\right)\right) \tag{4}$$

A major limitation of the original WeaveNet fairness formulation lies in its exclusive focus on the difference between the two sides' satisfaction levels, disregarding their absolute magnitudes. This design can lead to misleading fairness evaluations, as matchings where both sides are equally but poorly satisfied are treated as perfectly fair. For example, a matching with $S(\hat{m}; A) = 50$ and $S(\hat{m}; B) = 50$ yields the same fairness loss ($= 0$) as one with $S(\hat{m}; A) = 500$ and $S(\hat{m}; B) = 500$,

even though the latter is objectively superior for both sides. To address this issue, we propose a **logarithmic difference** formulation:

$$\mathcal{L}_f^{\text{log-dif}}(\hat{m}; I) = \frac{1}{ln(S(\hat{m}; A))} + \frac{1}{ln(S(\hat{m}; B))} + |S(\hat{m}; A) - S(\hat{m}; B)| \tag{5}$$

It explicitly penalizes low satisfaction scores through the logarithmic terms while maintaining the balance constraint via the absolute difference. The logarithmic terms ensure that matchings with higher overall satisfaction are preferred, while the difference term enforces fairness between sides. This formulation encourages the model to simultaneously maximize both sides' satisfaction and maintain balance, leading to outcomes that are both fair and of high quality. The logarithmic scale also provides stronger gradients when satisfaction scores are low, helping the model escape poor local minima during training.

In all GNN, GCN, and GAT models, we employ the same two loss components as defined in WeaveNet, the Matrix Constraint Loss ($\mathcal{L}_m$) and the Blocking Pair Suppression Loss ($\mathcal{L}_s$), while substituting WeaveNet's fairness term ($\mathcal{L}_f$) with one of our four proposed fairness loss functions. They can be flexibly interchanged depending on the experimental setup, by substitution into Equation (6). We compare their effectiveness empirically in Section 4.

**Full overall loss function with modified $\mathcal{L}_f$:**

$$\mathcal{L}_{\text{fsm}}(\hat{m}) = \lambda_m \mathcal{L}_m + \frac{1}{2} \sum_{m \in \{\hat{m}^A, \hat{m}^B\}} (\lambda_s \mathcal{L}_s(m) + \lambda_f \mathcal{L}_f(m)) \tag{6}$$

where $\hat{m}^A = \text{softmax}(\hat{m})$ and $\hat{m}^B = \text{softmax}(\hat{m}^\top)$.

## 3.1 Evaluation Metrics

The final metrics with which the matchings are evaluated are ESD, ARD, WLR, and EF-k, all defined below. We also evaluate on the metrics from Sone et al. [2023b]: SEq, Reg, Egal, Bal (See Table 1).
**Exponential Score Difference (ESD):** This metric captures the exponential growth of dissatisfaction by computing $e^{|M(u)-M(v)|}$, where $M(u)$ and $M(v)$ are the ranks of assigned colleges for two students. A higher ESD indicates a greater imbalance in satisfaction between students.
**Average Rank Difference (ARD):** Defined as $\frac{|M(u)-M(v)|}{|V_1|}$, ARD measures the average difference in rank between matched entities. This provides a normalized view of disparity across all pairs in the matching.
**Weighted Logarithmic Rank (WLR):** This function balances rank magnitude and fairness by computing $M(u)\ln(M(u)) + M(v)\ln(M(v)) + |M(u) - M(v)|$, which penalizes both high rank assignments and uneven distributions.
**Envy-Freeness** (EF-$k$): EF-$k$ assesses the fairness of a matching by checking whether any student envies another student who has a better assignment with higher priority. The smaller the value of $k$, the closer the matching is to being envy-free.
Our evaluation uses both WeaveNet's group-level fairness metrics (SEq, Reg, Egal, Bal) and individual-level fairness metrics (EF-k, ESD, ARD, WLR) to provide a comprehensive assessment of matching quality from both collective and individual perspectives.

## 4 Experiments

### 4.1 Experimental Setup

In our evaluation, we are focusing on Graph Neural Networks (GNN) and Graph Convolutional Networks (GCN). Each architecture is tested in two configurations: without an attention mechanism and with a final attention layer.

For the configurations **without attention**, we consider networks with 6, 16, and 32 layers. For the configurations **with attention**, we consider networks with 7, 17, and 33 layers, where the last layer implements the attention mechanism. This setup allows us to study both the effect of depth and the impact of adding an attention layer. The regularization parameters were set to the values used by Sone et al. [2023b]: $\lambda_m = 1.0$, $\lambda_s = 0.7$, $\lambda_f = 0.01$.

## 4.2 Evaluation Protocol

In our setup both of the architectures are learning using the losses defined in Section 3. Performance is assessed using the metrics defined in Section 3.1.We compute the losses for $N = 3, 5, 7$, and $9$, following the distributions described in Section4.3.

## 4.3 Data

In our experiments, we follow the same methodology as described in Tziavelis et al. [2019a]. We use synthetic datasets where user preferences are generated from four different distributions: **Uniform (UU)**, **Discrete (DD)**, **Gaussian (GG)** and a **Mixed (UD)** distribution combining uniform and discrete preferences. Additionally, we include a real-world dataset derived from user rating activity on the online dating platform **LibimSeTi (Lib)**, as described in Brozovsky and Petricek [2007].

## 4.4 Results

Our experimental results reveal that while the proposed architectures achieve *perfect stability* across all problem sizes (Tables 2–4), they fail to meaningfully optimize fairness objectives. Specifically, for all four fairness loss variants, the fairness component $L_f$ remains near zero when trained with $\lambda_f = 0.01$, whereas the stability term ($L_s = 0.022$) continues to dominate the optimization. This imbalance indicates that the model primarily learns to minimize blocking pairs, favoring stability, while effectively ignoring fairness. We interpret this not as a model deficiency, but as evidence of **implicit bias introduced by the training process itself**: small fairness weights can overpower architectural innovations and dictate the optimization trajectory.

Distribution-specific analyses (Table 3) further illustrate this behavior. Although uniform and mixed distributions yield moderate sex-equality (SEq $\approx 6.0$) and balanced satisfaction (Bal $\approx 36.0$), discrete and Gaussian preference structures exhibit disproportionately higher SEq ($\approx 27.0$ and $\approx 22.0$, respectively), suggesting that fairness degradation intensifies with more structured or skewed preference data. This pattern underscores how the interaction between data distribution and hyperparameterization can systematically bias the outcomes of fairness-aware learning.

## 5 Discussion and Limitations

Previous work, such as Sone et al. [2023b], suggests that Graph Convolutional Networks(GCN) are not suitable for biparite matching tasks. However, there are no reasoning behind these observations. In our work, with experiments we provide empirical evidence to elaborate on that claim.

Despite introducing four loss variants, our results reveal that with $\lambda_f = 0.01$, the model primarily optimizes for stability ($L_s = 0.022$) while largely ignoring fairness objectives ($L_f = 0.0000$ for squared/absolute). This demonstrates how hyperparameter choices can introduce implicit biases that override architectural design choices.

Additionally, our analysis is limited by the small-scale nature of the benchmark problems ($N \leq 9$), which may mask the severity of fairness-stability trade-offs that emerge in larger markets and our lightweight architecture approach. Unlike WeaveNet, which trains 200k steps with large batches on a deep, edge-aware architecture, our GNN/GCN+GAT runs used substantially fewer updates. Consequently, a small fairness weight ($\lambda_f = 0.01$) was insufficient to influence training, and larger $\lambda_f$ or longer training is required to move fairness metrics in our setting. The fairness term was effectively overshadowed by stronger stability objectives, leading the model to converge prematurely before fairness adjustments could take effect. Therefore, we suspect that a systematic grid search or adaptive reweighting could reveal more nuanced relationships between fairness and stability. Finally, while our models capture general distributional effects, real-world datasets (e.g., LibimSeTi) introduce noise and heterogeneity that warrant further exploration.

## 6 Conclusion

This work proposes a new logarithmic-difference fairness loss (Equation 5) that addresses a fundamental limitation in WeaveNet's formulation by explicitly rewarding high satisfaction levels

alongside balance. Our experiments demonstrate that GCN architectures can achieve perfect stability in small-scale matching problems ($N \leq 9$), but learning nuanced fairness objectives requires careful hyperparameter tuning, specifically, fairness loss weights must be sufficiently large ($\lambda_f > 0.01$) to impact training dynamics.

Our work highlights the importance of loss weight calibration as a source of implicit bias in fairness-aware optimization. While architectural innovations like attention mechanisms and specialized graph structures (WeaveNet) receive significant research attention, our results suggest that hyperparameter choices may have equally significant impacts on fairness outcomes.

Future work could: (1) investigate optimal $\lambda_f$ values through systematic grid search, (2) evaluate performance on larger problem instances ($N \geq 20$) where fairness-stability tradeoffs become more pronounced, and (3) develop adaptive loss weighting schemes that automatically balance multiple objectives during training.

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

Table 1: WeaveNet Fairness Cost Definitions

| Cost | Definition |
|---|---|
| Sex Equality | $SEq(m; I) = |P(m; A) - P(m; B)|$ |
| Regret | $Reg(m; I) = \max_{\{a_i, b_j\} \in m}(\max(p_{ij}^A, p_{ji}^B))$ |
| Egalitarian | $Egal(m; I) = P(m; A) + P(m; B)$ |
| Balance | $Bal(m; I) = \max(P(m; A), P(m; B))$ |

## A  Preliminaries

See Table 1, where:

$$P(m; A) = \sum_{\{a_i, b_j\} \in m} p_{ij}^A, \quad P(m; B) = \sum_{\{a_i, b_j\} \in m} p_{ij}^B.$$

## B  Results

Table 2: Performance comparison of GCN-6 across different loss variants. All models achieve 100% stability with 0 blocking pairs.

| Loss Variant | Sex Equality (SEq) ↓ | | | | Balance (Bal) ↓ | | | |
|---|---|---|---|---|---|---|---|---|
| | $N = 3$ | $N = 5$ | $N = 7$ | $N = 9$ | $N = 3$ | $N = 5$ | $N = 7$ | $N = 9$ |
| Squared ($L_f^{sq}$) | 1.64 | 4.22 | 8.67 | 13.66 | 6.68 | 16.94 | 31.79 | 51.04 |
| Absolute ($L_f^{abs}$) | 1.64 | 3.76 | 8.67 | 13.70 | 6.68 | 16.82 | 31.78 | 51.06 |
| Log-exp ($L_f^{exp}$) | 1.64 | 4.22 | 8.67 | 13.69 | 6.68 | 16.94 | 31.78 | 51.06 |
| Log-dif ($L_f^{log-dif}$) | **1.35** | **4.22** | **5.18** | **13.69** | **6.67** | **16.94** | **31.73** | **51.06** |

Table 3: Performance across different preference distributions at $N = 9$ for GCN-6 models. All achieve 100% stability.

| Distribution | Loss | Average Metrics | | | | Worst-Case Metrics | | | |
|---|---|---|---|---|---|---|---|---|---|
| | | SEq↓ | EF-k↓ | ARD↓ | WLR↓ | SEq↓ | EF-k↓ | ARD↓ | WLR↓ |
| UU (Uniform) | Squared | 5.94 | 0.00 | 3.08 | 36.26 | 23 | 0 | 5.00 | 41.45 |
| | Absolute | 6.07 | 0.00 | 3.10 | 36.13 | 23 | 0 | 5.00 | 41.45 |
| | Log-exp | 6.07 | 0.00 | 3.10 | 36.13 | 23 | 0 | 5.00 | 41.45 |
| | Log-dif | 6.07 | 0.00 | 3.10 | 36.13 | 23 | 0 | 5.00 | 41.45 |
| DD (Discrete) | Squared | 27.02 | 0.00 | 3.31 | 30.12 | 35 | 0 | 4.11 | 41.45 |
| | Absolute | 27.14 | 0.00 | 3.31 | 30.02 | 35 | 0 | 4.11 | 30.88 |
| | Log-exp | 27.14 | 0.00 | 3.31 | 30.02 | 35 | 0 | 4.11 | 30.88 |
| | Log-dif | 27.14 | 0.00 | 3.31 | 30.02 | 35 | 0 | 4.11 | 30.88 |
| GG (Gaussian) | Squared | 21.93 | 0.00 | 3.16 | 39.33 | 34 | 0 | 4.33 | 41.45 |
| | Absolute | 21.96 | 0.00 | 3.16 | 39.41 | 34 | 0 | 4.33 | 41.45 |
| | Log-exp | 22.00 | 0.00 | 3.16 | 39.43 | 34 | 0 | 4.33 | 41.45 |
| | Log-dif | 22.00 | 0.00 | 3.16 | 39.43 | 34 | 0 | 4.33 | 41.45 |
| UD (Mixed) | Squared | 6.57 | 0.00 | 2.97 | 35.89 | 22 | 0 | 5.00 | 41.45 |
| | Absolute | 6.57 | 0.00 | 2.97 | 35.88 | 22 | 0 | 5.00 | 41.45 |
| | Log-exp | 6.57 | 0.00 | 2.97 | 35.88 | 22 | 0 | 5.00 | 41.45 |
| | Log-dif | 6.57 | 0.00 | 2.97 | 35.88 | 22 | 0 | 5.00 | 41.45 |
| Lib (Real-world) | Squared | 6.73 | 0.00 | 2.92 | 35.75 | 26 | 0 | 5.00 | 41.45 |
| | Absolute | 6.66 | 0.00 | 2.93 | 35.73 | 26 | 0 | 5.00 | 41.45 |
| | Log-exp | 6.66 | 0.00 | 2.93 | 35.73 | 26 | 0 | 5.00 | 41.45 |
| | Log-dif | 6.66 | 0.00 | 2.93 | 35.73 | 26 | 0 | 5.00 | 41.45 |

Table 4: Detailed performance of GCN-6 with Log-dif loss ($L_f^{log-dif}$) across problem sizes and distributions.

| Distribution | Average SEq↓ | | | | Average Balance↓ | | | |
|---|---|---|---|---|---|---|---|---|
| | $N = 3$ | $N = 5$ | $N = 7$ | $N = 9$ | $N = 3$ | $N = 5$ | $N = 7$ | $N = 9$ |
| UU (Uniform) | 1.04 | 2.32 | 4.22 | 6.07 | 6.59 | 16.21 | 30.14 | 48.21 |
| DD (Discrete) | 1.52 | 7.54 | 7.06 | 27.14 | 7.52 | 15.00 | 34.42 | 45.00 |
| GG (Gaussian) | 1.92 | 6.22 | 5.73 | 22.00 | 6.07 | 21.21 | 33.32 | 67.00 |
| UD (Mixed) | 1.18 | 2.38 | 4.32 | 6.57 | 6.60 | 16.16 | 30.41 | 47.97 |
| Lib (Real-world) | 1.08 | 2.67 | 4.21 | 6.66 | 6.56 | 16.13 | 30.50 | 47.13 |
| **Overall** | **1.35** | **4.22** | **5.18** | **13.69** | **6.67** | **16.94** | **31.73** | **51.06** |

Table 5: Performance comparison of GCN-16 across different loss variants. All models achieve 100% stability with 0 blocking pairs.

| Loss Variant | Sex Equality (SEq)↓ | | | | Balance (Bal)↓ | | | |
|---|---|---|---|---|---|---|---|---|
| | $N = 3$ | $N = 5$ | $N = 7$ | $N = 9$ | $N = 3$ | $N = 5$ | $N = 7$ | $N = 9$ |
| Squared ($\mathcal{L}_f^{sq}$) | 1.64 | 3.66 | 5.19 | 11.41 | 6.68 | 16.41 | 31.71 | 49.20 |
| Absolute ($\mathcal{L}_f^{abs}$) | 1.64 | 2.96 | 8.64 | 13.69 | 6.68 | 16.87 | 31.79 | 51.06 |
| Log-exp ($\mathcal{L}_f^{exp}$) | 1.64 | 4.22 | 5.78 | 13.69 | 6.68 | 16.94 | 30.67 | 51.06 |
| Log-dif ($\mathcal{L}_f^{log-dif}$) | **1.64** | **4.22** | **8.67** | **13.69** | **6.68** | **16.94** | **31.78** | **51.06** |

Table 6: Performance across different preference distributions at $N = 9$ for GCN-16 models. All achieve 100% stability.

| Distribution | Loss | Average Metrics | | | | Worst-Case Metrics | | | |
|---|---|---|---|---|---|---|---|---|---|
| | | SEq ↓ | EF-k ↓ | ARD ↓ | WLR ↓ | SEq ↓ | EF-k ↓ | ARD ↓ | WLR ↓ |
| UU (Uniform) | Squared | 6.95 | 0.00 | 2.98 | 35.69 | 22 | 0 | 4.56 | 41.45 |
| | Absolute | 6.07 | 0.00 | 3.10 | 36.13 | 23 | 0 | 5.00 | 41.45 |
| | Log-exp | 6.07 | 0.00 | 3.10 | 36.13 | 23 | 0 | 5.00 | 41.45 |
| | Log-dif | 6.07 | 0.00 | 3.10 | 36.13 | 23 | 0 | 5.00 | 41.45 |
| DD (Discrete) | Squared | 27.00 | 0.00 | 3.29 | 30.10 | 34 | 0 | 4.00 | 30.88 |
| | Absolute | 27.14 | 0.00 | 3.31 | 30.02 | 35 | 0 | 4.11 | 30.88 |
| | Log-exp | 27.14 | 0.00 | 3.31 | 30.02 | 35 | 0 | 4.11 | 30.88 |
| | Log-dif | 27.14 | 0.00 | 3.31 | 30.02 | 35 | 0 | 4.11 | 30.88 |
| GG (Gaussian) | Squared | 10.64 | 0.00 | 2.86 | 37.41 | 25 | 0 | 5.00 | 41.45 |
| | Absolute | 22.00 | 0.00 | 3.16 | 39.41 | 34 | 0 | 4.33 | 41.45 |
| | Log-exp | 22.00 | 0.00 | 3.16 | 39.43 | 34 | 0 | 4.33 | 41.45 |
| | Log-dif | 22.00 | 0.00 | 3.16 | 39.43 | 34 | 0 | 4.33 | 41.45 |
| UD (Mixed) | Squared | 5.92 | 0.00 | 2.98 | 36.11 | 21 | 0 | 5.11 | 41.45 |
| | Absolute | 6.57 | 0.00 | 2.97 | 35.88 | 22 | 0 | 5.00 | 41.45 |
| | Log-exp | 6.57 | 0.00 | 2.97 | 35.88 | 22 | 0 | 5.00 | 41.45 |
| | Log-dif | 6.57 | 0.00 | 2.97 | 35.88 | 22 | 0 | 5.00 | 41.45 |
| Lib (Real-world) | Squared | 6.49 | 0.00 | 2.92 | 36.47 | 21 | 0 | 4.67 | 41.45 |
| | Absolute | 6.66 | 0.00 | 2.93 | 35.73 | 26 | 0 | 5.00 | 41.45 |
| | Log-exp | 6.66 | 0.00 | 2.93 | 35.73 | 26 | 0 | 5.00 | 41.45 |
| | Log-dif | 6.66 | 0.00 | 2.93 | 35.73 | 26 | 0 | 5.00 | 41.45 |

Table 7: Detailed performance of GCN-16 with Log-dif loss ($\mathcal{L}_f^{log-dif}$) across problem sizes and distributions.

| Distribution | Average SEq ↓ | | | | Average Balance ↓ | | | |
|---|---|---|---|---|---|---|---|---|
| | $N = 3$ | $N = 5$ | $N = 7$ | $N = 9$ | $N = 3$ | $N = 5$ | $N = 7$ | $N = 9$ |
| UU (Uniform) | 1.09 | 2.32 | 4.81 | 6.07 | 6.48 | 16.21 | 30.63 | 48.21 |
| DD (Discrete) | 3.00 | 7.54 | 17.56 | 27.14 | 6.00 | 15.00 | 28.00 | 45.00 |
| GG (Gaussian) | 1.86 | 6.22 | 12.77 | 22.00 | 7.83 | 21.21 | 40.76 | 67.00 |
| UD (Mixed) | 1.15 | 2.38 | 4.40 | 6.57 | 6.56 | 16.16 | 30.16 | 47.97 |
| Lib (Real-world) | 1.13 | 2.67 | 3.80 | 6.66 | 6.54 | 16.13 | 29.37 | 47.13 |
| **Overall** | **1.64** | **4.22** | **8.67** | **13.69** | **6.68** | **16.94** | **31.78** | **51.06** |

Table 8: Performance across different preference distributions at $N = 9$ for GNN-16 models. All achieve 100% stability.

| Distribution | Loss | Average Metrics | | | | Worst-Case Metrics | | | |
|---|---|---|---|---|---|---|---|---|---|
| | | SEq ↓ | EF-k ↓ | ARD ↓ | WLR ↓ | SEq ↓ | EF-k ↓ | ARD ↓ | WLR ↓ |
| UU (Uniform) | Squared | 6.37 | 0 | 2.93 | 35.98 | 22 | 0 | 5.22 | 41.45 |
| | Absolute | 6.37 | 0 | 2.93 | 35.98 | 22 | 0 | 5.22 | 41.45 |
| | Log-exp | 6.37 | 0 | 2.93 | 35.98 | 22 | 0 | 5.22 | 41.45 |
| | Log-dif | 6.37 | 0 | 2.93 | 35.98 | 22 | 0 | 5.22 | 41.45 |
| DD (Discrete) | Squared | 27.02 | 0 | 3.30 | 30.08 | 35 | 0 | 4.11 | 30.88 |
| | Absolute | 27.02 | 0 | 3.30 | 30.08 | 35 | 0 | 4.11 | 30.88 |
| | Log-exp | 27.02 | 0 | 3.30 | 30.08 | 35 | 0 | 4.11 | 30.88 |
| | Log-dif | 27.02 | 0 | 3.30 | 30.08 | 35 | 0 | 4.11 | 30.88 |
| GG (Gaussian) | Squared | 20.85 | 0 | 3.21 | 39.16 | 36 | 0 | 5.00 | 41.45 |
| | Absolute | 20.85 | 0 | 3.21 | 39.16 | 36 | 0 | 5.00 | 41.45 |
| | Log-exp | 20.85 | 0 | 3.21 | 39.16 | 36 | 0 | 5.00 | 41.45 |
| | Log-dif | 20.85 | 0 | 3.21 | 39.16 | 36 | 0 | 5.00 | 41.45 |
| UD (Mixed) | Squared | 6.18 | 0 | 2.95 | 36.24 | 28 | 0 | 5.11 | 41.45 |
| | Absolute | 6.18 | 0 | 2.95 | 36.24 | 28 | 0 | 5.11 | 41.45 |
| | Log-exp | 6.18 | 0 | 2.95 | 36.24 | 28 | 0 | 5.11 | 41.45 |
| | Log-dif | 6.18 | 0 | 2.95 | 36.24 | 28 | 0 | 5.11 | 41.45 |
| Lib (Real-world) | Squared | 6.27 | 0 | 2.94 | 35.94 | 28 | 0 | 5.00 | 41.45 |
| | Absolute | 6.27 | 0 | 2.94 | 35.94 | 28 | 0 | 5.00 | 41.45 |
| | Log-exp | 6.27 | 0 | 2.94 | 35.94 | 28 | 0 | 5.00 | 41.45 |
| | Log-dif | 6.27 | 0 | 2.94 | 35.94 | 28 | 0 | 5.00 | 41.45 |

