# OpenReview forum: "On Fair and Balanced Matching in Bipartite Graphs"
_EurIPS.cc/2025/Workshop/UPLB — UPLB2025_

### Official Review · Reviewer_WjPf · 2025-10-25
**An interesting problem, a possibly incomplete analysis**

**Rating:** 3
**Confidence:** 3

**Review:**

The paper explores the effect of suitably designed loss function to train neural networks in providing stable and fair matchings on bipartite graphs. The problem is of course timely and interesting. However, my impression is that the submission is at a too early stage for publication and in a form that makes it not very clear. Here below some comments:
- The different contributions to $\mathcal L_{\rm fsm}$ appearing in Eq. 6 are undefined. Also, what is the meaning of the sum? From the previous definitions it looks like $\mathcal L_{\rm f}$ can take only an $N\times M$ matrix, and not its transpose in general. Please correct me if I am wrong.
- The proposed loss sounds compatible with the goal of the task, but there is no much theoretical justification and one might wonder why it is preferable to other constructions involving functions monotonically decreasing with $|S|$ (e.g., a contribution like $\exp(-|S(\hat m;A)|^2-|S(\hat m;B)|^2)$, just to make an example).
- In Section 3.1, there is a sudden reference to "college datasets" and other quantities like $M(u)$: a connection with the prior discussion is lacking. It seems the authors have in mind some specific dataset and task, that the reader has not been introduced to. Also, I suppose the metrics are intended as averaged over all pairs. The relation with the actual adopted dataset discussed in Section 4.3 is unclear.
- The experiments are performed on various architectures (GNN, GCN) with and without attention. However the conclusion are given for a precise choice of the hyperparameters only: such a choice is arbitrary and, as the authors seem to be aware of, might be crucial for the robustness of the results themselves. In other words, the apparent insensitivity of the training to the fairness portion of the loss might well be related to the specific choice of such hyperparameters. Without doing an extensive search as proposed by the authors, a collection of a limited number of tests would have possibly given some hints with this respect. Another limitation, acknowledged by the authors, is the very small scale of the benchmark problems ($N\leq 9$).

Summarising, the work investigates the failure of such architectures in keeping into account a fairness contribution in a specific loss with specific parameters and small problem sizes, so that it is unclear if any conclusion can be drawn from it for the general case. It is not also clear to me (possibly, at higher level) what is (if any) the benefit of using neural networks with respect to any "classical approach" aiming at the simple minimization of $\mathcal L_{\rm fsm}$. Overall, the authors themselves seem to be aware of the fact that the limitations of the analysis are such that, even by restricting to the discussed setup, it is possible that the fairness metric did not play a role because of limited training updates and/or problem sizes (see lines 152-154). Although, as anticipated, the problem is interesting, my impression is that the current manuscript needs a revision and extension before publication.

Side remark — Citations are given as "in-text" and are sometimes confusing (eg, instead of ``Gale and Shapley [1962]``, it would be better to use something like ``[Gale and Shapley, 1962]``. In section 4 and 5 there is a number of references to $L$-quantities that are likely supposed to be read as $\mathcal L$-quantities.

---

### Decision · Program_Chairs · 2025-11-03

Accept (Poster)